# Validation of the severe COVID-19 prognostic value of serum IL-6, IFN-λ3, CCL17, and calprotectin considering the timing of clinical need for prediction

**Kei Yamamoto**[1]*, **Yusuke Ohsiro**[2], **Tetsuya Suzuki**[1], **Michiyo Suzuki**[1], **Sayaka Miura**[1], **Maki Nagashima**[1], **Noriko Iwamoto**[1], **Junko S. Takeuchi**[3], **Moto Kimura**[3], **Wataru Sugiura**[3], **Satoru Nebuya**[4], **Masato Kurokawa**[4], **Norio Ohmagari**[1]

**1** Disease Control and Prevention Center, National Center for Global Health and Medicine, Shinjuku-ku, Tokyo, Japan, **2** Clinical Laboratory, National Center for Global Health and Medicine, Shinjuku-ku, Tokyo, Japan, **3** Center for Clinical Sciences, National Center for Global Health and Medicine, Shinjuku-ku, Tokyo, Japan, **4** Sanyo Chemical Industries, Ltd., Higashiyama-ku, Kyoto, Japan

* kyamamoto@hosp.ncgm.go.jp

**Data Availability Statement:** All relevant data are within the paper and its Supporting Information files.

## Abstract

Although biomarkers to predict coronavirus disease 2019 (COVID-19) severity have been studied since the early pandemic, no clear guidelines on using them in clinical practice are available. Here, we examined the ability of four biomarkers to predict disease severity using conserved sera from COVID-19 patients who received inpatient care between January 1, 2020 and September 21, 2021 at the National Center for Global Health and Medicine, collected at the appropriate time for prediction. We predicted illness severity in two situations: 1) prediction of future oxygen administration for patients without oxygen support within 8 days of onset (Study 1) and 2) prediction of future mechanical ventilation support (excluding non-invasive positive pressure ventilation) or death of patients within 4 days of the start of oxygen administration (Study 2). Interleukin-6, IFN-λ3, thymus and activation-regulated chemokine, and calprotectin were measured retrospectively. Other laboratory and clinical information were collected from medical records. AUCs were calculated from ROC curves and compared for the predictive ability of the four biomarkers. Study 1 included 18 patients, five of whom had developed oxygen needs. Study 2 included 45 patients, 13 of whom required ventilator management or died. In Study 1, IFN-λ3 showed a good predictive ability with an AUC of 0.92 (95% CI 0.76–1.00). In Study 2, the AUC of each biomarker was 0.70–0.74. The number of biomarkers above the cutoff showed the possibility of good prediction with an AUC of 0.86 (95% CI 0.75–0.97). When two or more biomarkers were positive, sensitivity and specificity were 0.92 and 0.63, respectively. In terms of biomarker testing at times when prognostication may be clinically useful, IFN-λ3 was predictive of oxygenation demand and a combination of the four biomarkers was predictive of mechanical ventilator requirement.

**Funding:** K.Y. received research grants from Sanyo Chemical Industries, Ltd. [no grant number] for the submitted work. S. N. and M. Kurokawa. are employees of Sanyo Chemical Industries, Ltd. K.Y. received research grants from Fujirebio, Inc., Mizuho Medy, Co., Ltd., VisGene, Co., Ltd., Canon medical systems Co., and CarbGeM Inc., and M. Kimura received research grants from SB Coronavirus Inspection Center Corp., Canon medical systems Co., and Ezaki Glico Co., Ltd., and W. S received research grants from SB Coronavirus Inspection Center Corp. and Nippon genetics Co., Ltd. outside the submitted work. Sanyo Chemical Industries, Ltd. had a role in the conduct of the research and preparation of the manuscript. The funders had no additional role in study design, data collection and analysis, decision to publish, or preparation of the manuscript. The specific roles of these authors are articulated in the 'author contributions' section.

**Competing interests:** K.Y. received research grants from Sanyo Chemical Industries, Ltd. [no grant number] for the submitted work. S. N. and M. Kurokawa. are employees of Sanyo Chemical Industries, Ltd. K.Y. received research grants from Fujirebio, Inc., Mizuho Medy, Co., Ltd., VisGene, Co., Ltd., Canon medical systems Co., and CarbGeM Inc., and M. Kimura received research grants from SB Coronavirus Inspection Center Corp., Canon medical systems Co., and Ezaki Glico Co., Ltd., and W. S received research grants from SB Coronavirus Inspection Center Corp. and Nippon genetics Co., Ltd. outside the submitted work. This does not alter our adherence to PLOS ONE policies on sharing data and materials. K. Y. and Sanyo Chemical Industries have a patent pending on the calprotectin assay. There are no additional patents, products in development or marketed products associated with this research to declare.

## Introduction

Coronavirus disease 2019 (COVID-19) remains a worldwide pandemic. Because its severity varies, which sometimes results in rapid deterioration of respiratory status [1], accurate prediction is useful to select treatment sites. Not only anti-inflammatory therapy, such as corticosteroids, Janus kinase inhibitors, and interleukin (IL)-6 inhibitors, for moderate to critical COVID-19 cases, but also treatment of mild cases with monoclonal antibodies and oral antiviral drugs has been established. Therefore, although identification of prognostic biomarkers to classify a COVID-19 patient's current and future status is useful for optimal treatment, the use of prognostic biomarkers, which have been investigated since the beginning of the pandemic, has not been established. In addition to IL-6, which is associated with the risk of severe disease and death [2–11], although thymus and activation-regulated chemokine (TARC) and IFN-λ3 have been approved and available in Japan [12], they are not yet used in clinical practice. Although other biomarkers to predict severity, such as calprotectin (CP), have been reported [13], their utility has not been conclusively determined. Prognostic scoring combining clinical findings, underlying disease, and commonly used biomarkers such as C-reactive protein (CRP) and lactate dehydrogenase (LDH) has been proposed [14–16]. Although they are evaluated at certain times, such as at the time of hospitalization, the timing of hospitalization is not uniform for COVID-19 patients and is not always the optimal time to predict the clinically optimal prognosis. Therefore, in this study, we used conserved sera from COVID-19 patients to predict exacerbations in two situations where prognostication may be useful in clinical practice. One was to predict whether oxygen administration will be required at the time of mild disease (Fig 1, prediction A) and the other was to determine whether oxygen administration was already required, but further support such as ventilator management would be needed (Fig 1, prediction B). In these situations, we examined the prognostic value of four biomarkers (IFN-λ3, TARC, IL-6, and CP) for COVID-19.

## Methods

### Study design

We conducted a retrospective single-center observational study of COVID-19 patients who were admitted to the National Center for Global Health and Medicine, Tokyo, Japan.

Study 1: We evaluated the predictive ability of each biomarker for future oxygen demand using serum samples collected within 8 days of disease onset between March 6, 2020 and March 14, 2021 from patients who had not been administered oxygen.

Study 2: We evaluated the predictive ability of each biomarker for future mechanical ventilator demand or death using serum samples collected within 4 days of oxygen administration between January 1, 2020 and September 21, 2021 from patients who had not needed a mechanical ventilator.

### Participants

This study enrolled hospitalized patients aged 20 years or older, who consented in writing to participate in the study and to the secondary use of clinical information and specimens in the "A PROSPECTIVE OBSERVATIONAL STUDY OF CORONAVIRUS DISEASE 2019 (COVID-2019)", which was approved by the ethical committee of the National Center for Global Health and Medicine (NCGM-G-003472). Patients managed by the Disease Control and Prevention Center, whose serum specimens were stored, were included in the study. The severity of patients was determined arbitrarily in a situation where it was already known.

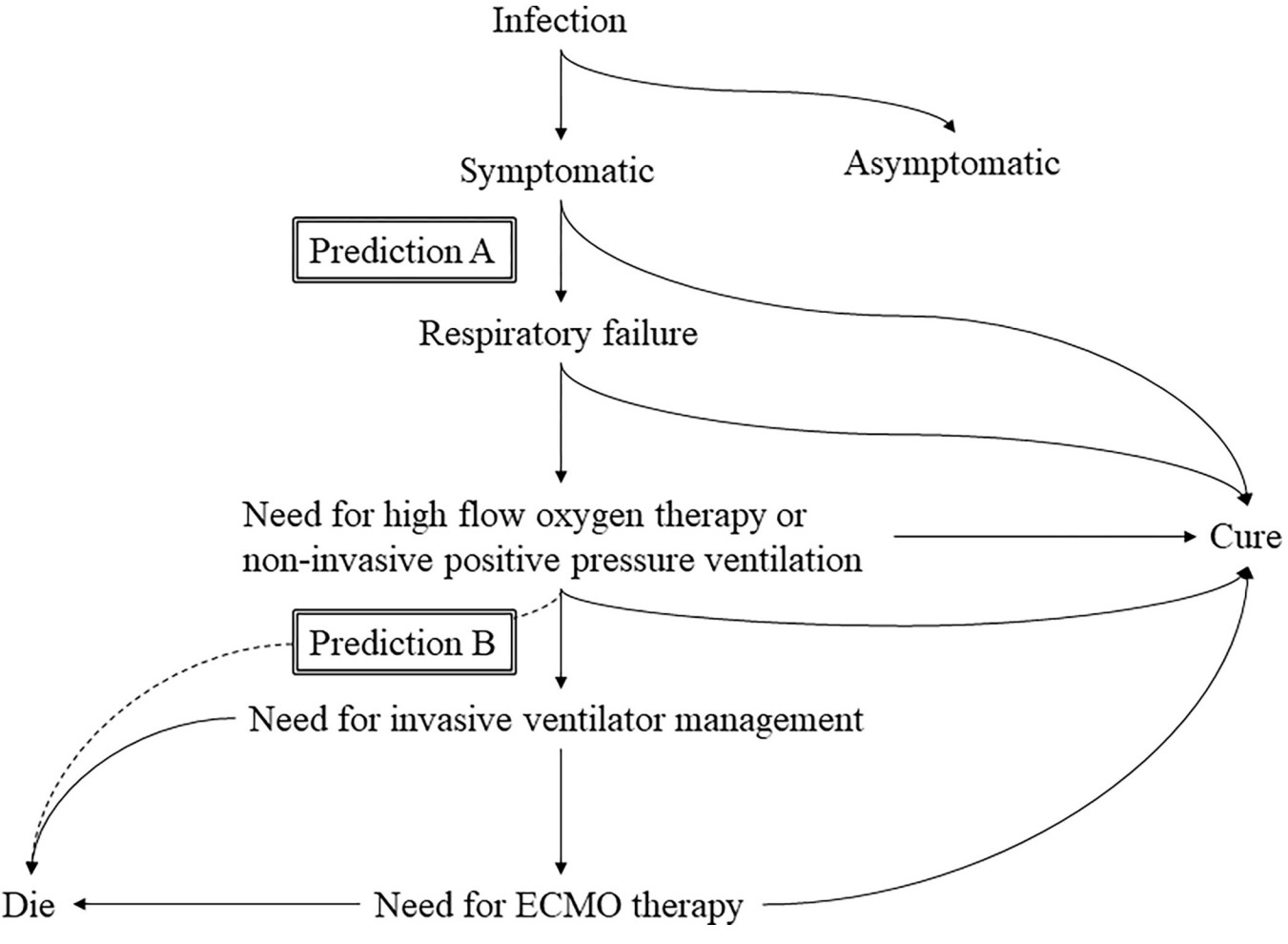

**Fig 1. Natural history of COVID-19 and the point at which prediction is needed.**

Information about the study, which advised that patient records might be used in medical research, was shown on both the hospital website and bulletin board with an opt-out participation option. This was substituted for participant consent. The study protocol, including the opt-out consent method, was approved by the Ethical Committee of the National Center for Global Health and Medicine (NCGM-G-004198-01 and NCGM-S-004361-02) on December 10, 2021 and February 14, 2022, respectively.

## Collection and definition of clinical information

The following information about patients was extracted from the hospital medical records: age, sex, underlying diseases (diabetes mellitus, malignancy, hypertension, cerebral vascular diseases, and congestive heart failure), height and weight (on admission), use of remdesivir, immunosuppressive agents (baricitinib and tocilizumab), and corticosteroids (e.g., dexamethasone.) at the time of specimen collection, symptoms present up to the time of admission (fever, cough, respiratory distress, wheezing, and malaise), presence of oxygen demand on the day of collection and oxygen device used, CRP if collected on the day of blood collection, and laboratory data on admission (lymphocyte count, LDH, and CRP).

We calculated the PREDI-CO score by the severity prediction model extracted from the above data [14]. The PREDI-CO score predicts severe respiratory failure, which is defined by WHO criteria $SpO_2$ <93% with 100% $FiO_2$ (reservoir mask, continuous positive airway pressure ventilation, or other non-invasive ventilation), respiratory rate >30 breaths/minute, or respiratory distress. Three or higher predicts a high risk for severe disease. The simple risk score by the oxygen administration prediction model created from registry data in Japan [17]. The model includes three age groups, 18–39, 40–64, and ≥65 years with ≥6, ≥5, and ≥3, respectively, predicting a high risk of receiving oxygen in the future. Symptoms with entries of 'unknown' were treated as missing values and missing data for symptoms were treated as no symptoms.

## Blood tests and serum samples

Conserved sera were refrigerated immediately after sampling and frozen at −80˚C for up to 7 days after sampling. The samples were thawed prior to study and labeled so that only the research identification code could be referenced by a research collaborator who was not responsible for the serum testing. CP and IFN-λ3 were measured by a collaborator at the National Center for Global Health and Medicine, and IL-6 and TARC were measured by an external laboratory (BML Inc., Tokyo, Japan). The methods of measurement of various biomarkers are shown in S1 Appendix.

## Outcomes

The primary outcome was the ability to predict the presence or absence of oxygen therapy demand in Study 1 and the presence or absence of mechanical ventilator demand in Study 2. As a secondary outcome of Study 2, we examined the predictive ability of each biomarker with the presence or absence of oxygen supply with a nasal high flow cannula (NHFC). The cutoff values for IFN-λ3 and TARC were set at ≥13.6 and ≤95 pg/mL, respectively, as per the package insert of each test kit. The cutoff for IL-6 was unclear and the cutoff to predict severe disease was set at ≥42 pg/mL based on median values in nine previous reports [2–10]. However, the optimal cutoff values for IFN-λ3, TARC, and IL-6 were calculated again by drawing ROC curves. If they differed significantly from the known cutoffs, the cutoffs were used for verification. The CP data were only from the enzyme-linked immunosorbent assay method and those from the absorption immunochromatography method were not used in the analysis. Data of CP (absorption immunochromatography) were used only to verify the correlation between absorption immunochromatography and in all test samples in S1 Fig.

## Statistical analysis

Qualitative and quantitative data are expressed as numbers (percentages) and medians [interquartile range (IQR)], respectively. Two-group comparisons were made with Fisher's exact test for discrete data and the Mann–Whitney U-test for continuous data. Receiver operating characteristic (ROC) curves were drawn for the predictive ability of each biomarker for primary or secondary outcomes and AUCs (95% confidence interval) were calculated. Optimal cutoffs were calculated from the ROC curve using Youden's index. For the four biomarkers and two predictive scores, the sensitivity, specificity, positive predictive value, and negative predictive value were calculated using known cutoffs (if established) and optimal cutoffs. The AUCs calculated for each biomarker were compared by the Delong method and corrected by the Holm method for multiple comparisons. Significance probability was 0.05. Statistical analysis was performed using EZR for Windows version 1.54 [18].

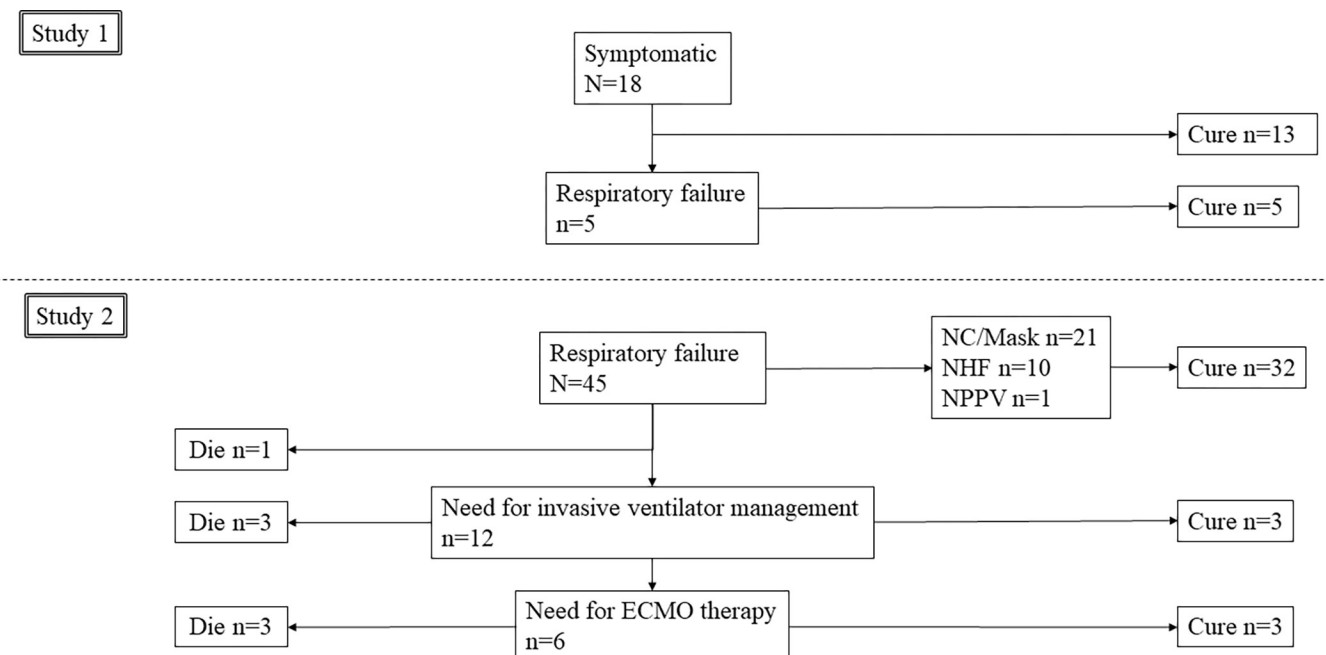

**Fig 2. Clinical outcomes of eligible patients.** NC: nasal canula, NHFC: Nasal high-flow cannula, NPPV: Non-invasive positive pressure ventilation, ECMO: Extracorporeal membrane oxygenation.

## Results

### Study 1

Eighteen patients (11 males) were included in the study, five of whom had developed oxygen demand after a median of 4 days. Of the patients with oxygen demand, none required NHFC or mechanical ventilator management. No patient had died while hospitalized (Fig 2). Table 1 shows the patient characteristics.

The median values of IL-6, IFN-λ3, TARC, and CP were 10.4 (IQR 4.0–20.9) pg/mL, 4.5 (IQR 3.2–4.8) pg/mL, 140.3 (IQR 96.6–198.9) pg/mL, and 3.54 (IQR 2.0–5.0) μg/mL, respectively, in patients without future oxygen demand. In patients with future oxygen demand, the values were 26.3 (IQR 12.7–39.7) pg/mL, 9.5 (IQR 8.2–12.2) pg/mL, 161.0 (IQR 138.5–198.6) pg/mL, and 5.6 (IQR 2.9–5.7) μg/mL, respectively. ROC curves are shown in Fig 3A. The AUC values (95% CI) of IL-6, IFN-λ3, TARC, and CP were 0.76 (0.50–1.00), 0.92 (0.76–1.00), 0.42 (0.11–0.72), and 0.59 (0.23–0.72), respectively. In multiple comparisons, there was no significant difference in the AUC of each biomarker (S1A Table). The cutoff for CP was set at 5.6 pg/mL. The optimal cutoff values for IL-6, IFN-λ3, and TARC were 12.7, 8.1, and 139 pg/mL, respectively. These cutoff values were also used to calculate sensitivity and specificity (Table 2). The sensitivity and specificity of IFN-λ3 were also calculated using optimal cutoff values (Table 2).

CRP was measured in 15 patients with an AUC of 0.57 (95%CI: 0.18–0.96). The optimal cutoff was 4.3 mg/dL, and sensitivity and specificity are shown in Table 3.

### Study 2

Forty-five patients (31 males) were included in the study, of which 11 required NHFC, one required non-invasive positive pressure ventilation, and 12 required mechanical ventilator. Of these patients, six required ECMO therapy. Seven patients had died (15.6%) and one of them

**Table 1. Patient characteristics at serum sampling in Study 1.**

| | No oxygen demand | Oxygen demand | *P*-value |
|---|---|---|---|
| N | 13 | 5 | |
| Age, median years (IQR) | 46 (41–53) | 49 (43–57) | 0.52 |
| Male sex, n (%) | 10 (76.9%) | 1 (20.0%) | 0.047 |
| Body mass index, median kg/m² (IQR) | 22.7 (20.4–29.1) | 24.0 (23.0–24.2) | 0.92 |
| Day after symptom onset, median day (IQR) | 5 (3–5) | 6 (4–6) | 0.52 |
| Underlying diseases, n (%) | | | |
| Diabetes mellitus | 1 (7.7%) | 0 | 1.0 |
| Malignancy | 0 | 0 | NA |
| Remdesivir, n (%) | 0 | 0 | NA |
| Immunosuppressive agents, n (%) | 0 | 0 | NA |
| Corticosteroids, n (%) | 0 | 0 | NA |
| PREDI-CO score [13], median (IQR) | 1 (1–2) | 1 (0–1) | 0.15 |
| High risk*, n (%) | 1 (7.7%) | 0 | 1.0 |
| simple risk score [17], n: median (IQR) | | | |
| 18–39 years | n = 3: 6(3–6) | n = 1: 2 | |
| 40–64 years | n = 9: 4(3–5) | n = 3: 3 (0–3) | |
| ≥65 years | n = 1: 3 | n = 1: 12 | |
| High risk**, n (%) | High risk: 7 (53.8%) | High risk: 1 (20.0%) | 0.31 |

IQR: interquartile range, NA: not applicable.

*Three or higher points.

**Six or higher for 18–39 years of age; five or higher for 40–64 years if age; three or higher for ≥65 years of age.

did not receive mechanical ventilator management because of their background (Fig 2). In the case that led to mechanical ventilation, the situation occurred at a median of 2 days after serum sampling. Two patients were receiving an anti-IL-6 antibody (tocilizumab) and one patient was receiving a JAK inhibitor at the time of serum sampling. Table 3 shows the patient characteristics.

The PREDI-CO score was significantly higher in the oxygen demand group, and 14 (43.8%) cases without intubation and 9 (69.2%) cases with intubation or death were classified as high risk by the PREDI-CO score, but the difference was not statistically significant. IL-6, IFN-λ3, TARC, and CP were 38.5 (IQR 19.5–61.3) pg/mL, 12.2 (IQR 5.2–23.9) pg/mL, 99.9 (IQR 70.6–140.35) pg/mL, and 61.7 (IQR 46.2–86.2) μg/mL, respectively, in patients who did not require mechanical ventilator management. In cases that required mechanical ventilator support or death, the values were 76.0 (IQR 40.5–153.0) pg/mL, 24.6 (IQR 15.5–41.1) pg/mL, 61.7 (IQR 46.2–86.2) pg/mL, and 13.1 (IQR 10.3–29.4) μg/mL, respectively. The respective ROC curves are shown in Fig 3B. The AUC values (95% CI) of IL-6, IFN-λ3, TARC, and CP were 0.70 (0.52–0.88), 0.73 (0.56–0.90), 0.74 (0.58–0.91), and 0.71 (0.55–0.88), respectively. In multiple comparisons, there was no significant difference in the AUC of each biomarker (S1B Table).

The cutoff value for CP was set at 10.3 pg/mL. Optimal cutoff values for IFN-λ3 and TARC were 13.5 and 95.2, respectively, which were similar to the known cutoffs. Therefore, for these two biomarkers, the known cutoff values were used. The cutoff for IL-6 was calculated to be 76 pg/mL. Sensitivity, specificity, positive predictive value, and negative predictive value were calculated to predict the need for mechanical ventilator demand (Table 4). CRP was measured in all patients and the AUC (95% CI) was 0.68 (0.50–0.87). The optimal cutoff was calculated to be 13.8 mg/dL as shown in Table 4.

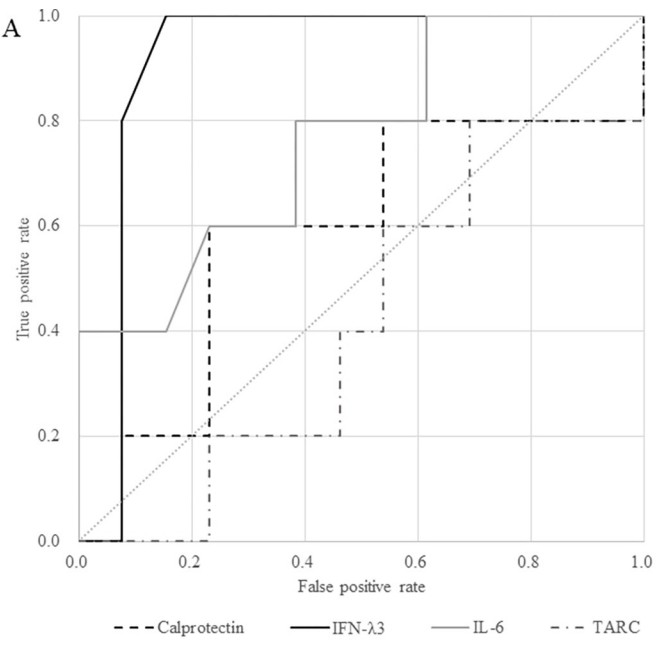

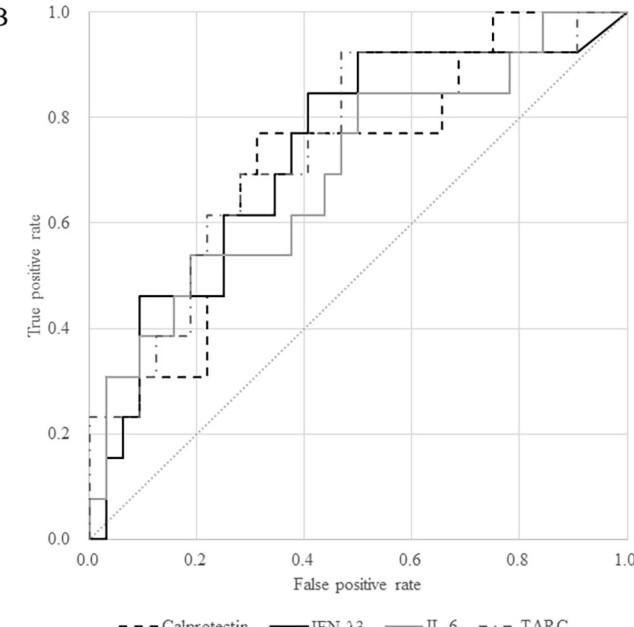

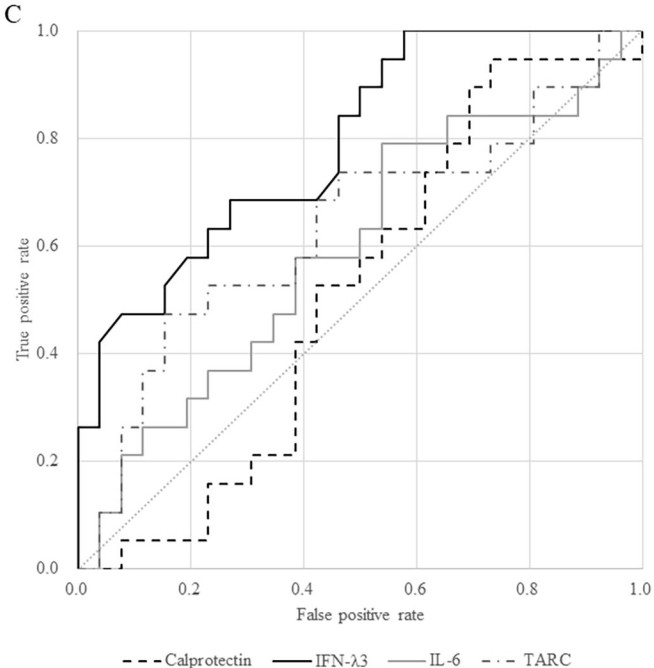

**Fig 3. Comparison of the predictive ability of biomarkers for disease severity in hospitalized individuals with a COVID-19 diagnosis.** A: Predictive ability for future oxygen administration in patients within 8 days after symptom onset, who did not require oxygen supply at the time of serum collection (Study 1). B: Predictive ability for future mechanical ventilation administration or death in patients with oxygen support except mechanical ventilation within 4 days after starting the oxygen supply (Study 2). C: Predictive ability for future oxygen support by a nasal high-flow cannula, non-invasive positive pressure ventilation, or mechanical ventilation in patients with oxygen support by a nasal canula or mask within 4 days after starting the oxygen supply (Study 2).

Using the scoring of the number of biomarkers positive at the cutoff calculated by the ROC curve for IL-6, IFN-λ3, TARC, and CP, predicting future mechanical ventilation demand was evaluated to have an AUC of 0.86 (95% CI: 0.75–0.97). When two or more biomarkers were positive, the results were better: 0.92 sensitivity (95% CI: 0.64–1.00), 0.63 specificity (95% CI:

**Table 2. Predictive ability of each biomarker and prediction score for future oxygen administration.**

| | Sensitivity | Specificity | Positive predictive value | Negative predictive value |
|---|---|---|---|---|
| Calprotectin (95% CI) | 0.60 (0.15–0.95) | 0.83 (0.52–0.98) | 0.60 (0.15–0.95) | 0.83 (0.52–0.98) |
| Cutoff value: ≥5.6 (μg/mL) | | | | |
| IFN-λ3 (95% CI) | 0.20 (0.01–0.72) | 0.92 (0.64–1.00) | 0.50 (0.01–0.99) | 0.75 (0.48–0.93) |
| Known cutoff | | | | |
| IFN-λ3 (95% CI) | 1.00 (0.36–1.00) | 0.85 (0.55–0.98) | 0.71 (0.29–0.96) | 1.00 (0.62–1.00) |
| Cutoff value: ≥8.1 (pg/mL) | | | | |
| Thymus and activation-regulated chemokine (95% CI) | 0.20 (0.01–0.72) | 0.77 (0.46–0.95) | 0.25 (0.01–0.81) | 0.71 (0.42–0.92) |
| Known cutoff | | | | |
| Thymus and activation-regulated chemokine (95% CI) | 0.40 (0.05–0.85) | 0.54 (0.25–0.81) | 0.25 (0.03–0.65) | 0.70 (0.35–0.93) |
| Cutoff value: ≤139 (pg/mL) | | | | |
| IL-6 (95% CI) | 0.20 (0.01–0.72) | 1.00 (0.66–1.00) | 1.00 (0.01–1.00) | 0.77 (0.50–0.93) |
| Known cutoff | | | | |
| IL-6 (95% CI) | 0.80 (0.28–1.00) | 0.62 (0.32–0.86) | 0.44 (0.14–0.79) | 0.89 (0.52–1.00) |
| Cutoff value: ≥12.7 (pg/mL) | | | | |
| C-reactive protein (95% CI) | 0.25 (0.01–0.81) | 0.82 (0.48–0.98) | 0.33 (0.01–0.91) | 0.75 (0.43–0.95) |
| Cutoff value: ≥4.3 (mg/dL) | | | | |
| PREDI-CO score [13] (95% CI) | 0 (0–0.64) | 0.92 (0.64–1.00) | 0 (0–0.99) | 0.71 (0.44–0.90) |
| Known cutoff | | | | |
| simple score [17] (95% CI) | 0.20 (0.01–0.72) | 0.46 (0.19–0.75) | 0.13 (0.003–0.53) | 0.60 (0.26–0.89) |
| Known cutoff | | | | |

CI: confidence interval.

0.44–0.79), 0.50 positive predictive value (95% CI: 0.29–0.71), and 0.95 negative predictive value (95% CI: 0.76–1.00) (Fig 4).

Of the 40 patients tested in situations that did not require a respiratory device above NHFC, in cases not requiring NHFC (n = 21), the median (IQR) values for IL-6, IFN-λ3, TARC, and CP were 40.2 (22.3–62.8) pg/mL, 10.9 (4.6–21.4) pg/mL, 103.5 (80.2–138.3) pg/mL, and 7.5 (4.9–10.4) μg/mL, respectively. In cases that required NHFC or higher respiratory devices or death (n = 19), the values were 48.8 (IQR 35.5–114.0) pg/mL, 25.5 (IQR 12.9–44.2) pg/mL, 67.5 (IQR 46.2–123.3) pg/mL, and 9.9 (IQR 6.4–13.1) μg/mL, respectively, with significant differences only for IFN-λ3 (p = 0.0017). AUC values (95% CI) of IL-6, IFN-λ3, TARC, and CP were 0.62 (0.44–0.80), 0.79 (0.66–0.93), 0.68 (0.50–0.85), and 0.61 (0.43–0.79), respectively (Fig 3C). Multiple comparisons of each AUC showed no significant difference between biomarkers (S1C Table). The median PREDI-CO score (IQR) was 2 (2–4) vs. 4 (2–4), which was slightly higher in the group with devices above NHFC, but not significantly. Sensitivity, specificity, positive predictive value, and negative predictive value (95% CI) of IFN-λ3 to predict requiring NHFC or higher oxygen support were 0.68 (0.43–0.87), 0.62 (0.38–0.82), 0.62 (0.38–0.82), and 0.68 (0.43–0.87), respectively.

## Discussion

In this study, we did not identify any biomarkers that were significantly better at predicting future oxygen administration in the absence of oxygen, future mechanical ventilator management, or death after oxygen administration had begun. However, IFN-λ3 performed well in the former when the cutoff was lowered and was the only biomarker that significantly differed between oxygen-treated and non-oxygen-treated patients. The prediction of mechanical

**Table 3. Patient characteristics at serum sampling in Study 2.**

| | Mechanical ventilation not required | Mechanical ventilation required or died | P-value |
|---|---|---|---|
| N | 32 | 13 | |
| Age, median years (IQR) | 65 (48–72) | 62 (56–68) | 0.93 |
| Male, n (%) | 21 (65.6%) | 10 (76.9%) | 0.72 |
| Body mass index, median kg/m$^2$ (IQR) | 26.3 (22.4–29.7) | 28.3 (23.5–32.2) | 0.61 |
| Day after symptom onset, median day (IQR) | 8 (6–9) | 7 (5–9) | 0.73 |
| Oxygen device used at sampling, n (%) | | | |
| Nasal canula/mask | 31 (96.9%) | 9 (69.2%) | 0.020 |
| NHFC | 1 (3.1%) | 2 (15.4%) | 0.20 |
| NPPV | 0 (0%) | 2 (15.4%) | 0.079 |
| Underlying diseases, n (%) | | | |
| Diabetes mellitus | 11 (34.4%) | 3 (23.1%) | 0.72 |
| Malignancy | 1 (3.1%) | 1 (7.7%) | 0.50 |
| Remdesivir, n (%) | 16 (50.0%) | 3 (23.1%) | 0.18 |
| Immunosuppressive agents, n (%) | 1 (3.1%) | 2 (15.4%) | 0.20 |
| Corticosteroids, n (%) | 19 (59.4%) | 9 (69.2%) | 0.74 |
| PREDI-CO score [13], median (IQR) | 3 (2–4) | 4 (3–5) | 0.019 |
| High risk*, n (%) | 14 (43.8%) | 9 (69.2%) | 0.19 |
| simple risk score [17], n: median (IQR) | | | |
| 18–39 years | n = 6: 6 (4–7) | n = 1: 5 | |
| 40–64 years | n = 10: 9 (6–10) | n = 7: 10 (6–10) | |
| ≥65 years | n = 16: 8 (5–12) | n = 5: 10 (9–10) | |
| High risk**, n (%) | 26 (81.3%) | 12 (92.3%) | 0.65 |

IQR: interquartile range, NHFC: nasal high flow cannula, NPPV: non-invasive positive pressure ventilation.

*Three or higher points.

**Six or higher for 18–39 years of age; five or higher for 40–64 years of age; three or higher for ≥65 years of age.

**Table 4. Predictive ability of each biomarker and prediction score for future mechanical ventilation.**

| | Sensitivity | Specificity | Positive predictive value | Negative predictive value |
|---|---|---|---|---|
| Calprotectin (95% CI) | 0.77 (0.46–0.95) | 0.69 (0.50–0.84) | 0.50 (0.27–0.73) | 0.88 (0.69–0.98) |
| Cutoff value: ≥10.3 (μg/mL) | | | | |
| IFN-λ3 (95% CI) | 0.77 (0.46–0.95) | 0.59 (0.41–0.76) | 0.44 (0.23–0.66) | 0.86 (0.65–0.97) |
| Known cutoff | | | | |
| Thymus and activation-regulated chemokine (95% CI) | 0.85 (0.55–0.98) | 0.53 (0.35–0.71) | 0.42 (0.23–0.63) | 0.90 (0.67–0.99) |
| Known cutoff | | | | |
| IL-6 (95% CI) | 0.69 (0.39–0.91) | 0.56 (0.38–0.74) | 0.39 (0.20–0.62) | 0.82 (0.60–0.95) |
| Known cutoff | | | | |
| IL-6 (95% CI) | 0.54 (0.25–0.81) | 0.81 (0.64–0.93) | 0.54 (0.25–0.81) | 0.81 (0.64–0.93) |
| Cutoff value: ≥76.0 (pg/mL) | | | | |
| C-reactive protein (95% CI) | 0.39 (0.14–0.68) | 0.88 (0.71–0.97) | 0.56 (0.21–0.86) | 0.78 (0.61–0.90) |
| Cutoff value: ≥13.8 (mg/dL) | | | | |
| PREDI-CO score [13] | 0.69 (0.39–0.91) | 0.56 (0.38–0.74) | 0.39 (0.20–0.62) | 0.82 (0.60–0.95) |
| Known cutoff | | | | |
| simple score [17] | 0.92 (0.64–1.00) | 0.19 (0.07–0.36) | 0.32 (0.18–0.49) | 0.86 (0.42–1.00) |
| Known cutoff | | | | |

CI: confidence interval.

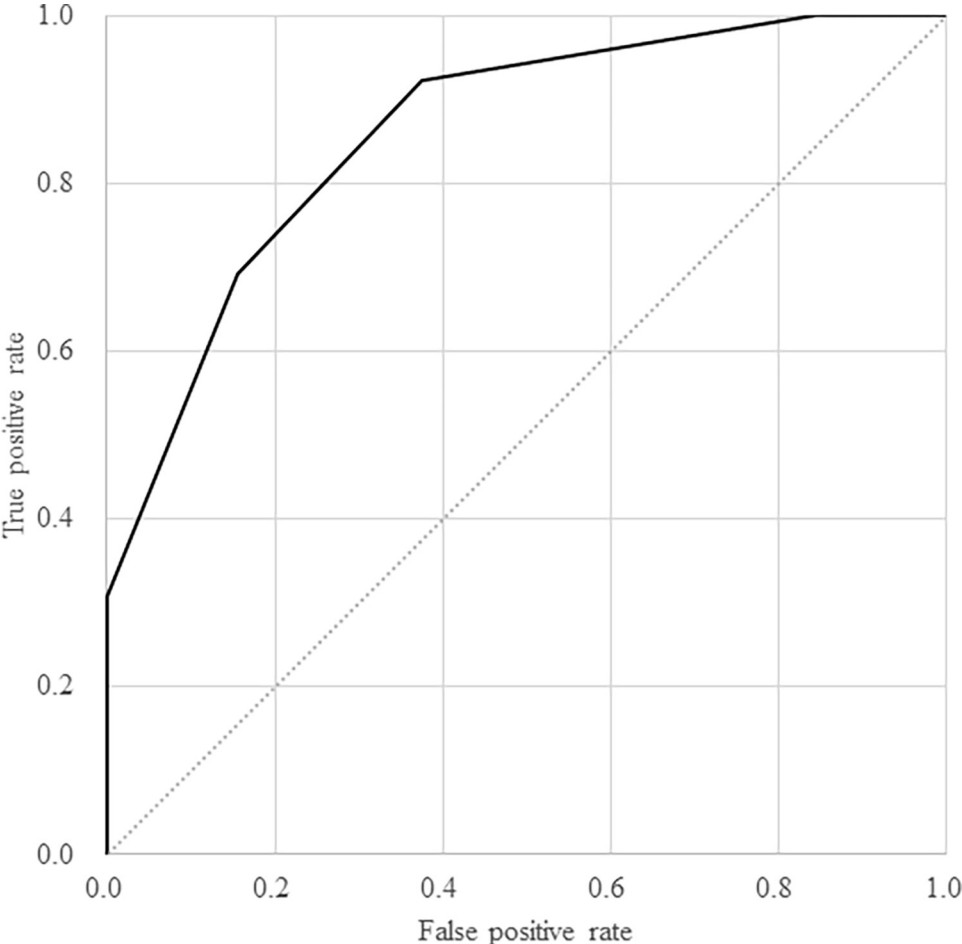

**Fig 4. Predictive ability for mechanical ventilation administration or death in patients with oxygen support except mechanical ventilation within 4 days after starting the oxygen supply by the number of positive results of the four biomarkers (Study 2).**

ventilator management was similar for IL-6, IFN-λ3, TARC, and CP biomarkers, and the PRE-DI-CO score.

IFN-λ is an antiviral cytokine that restricts pathogen infection and dissemination at barrier surfaces [19]. It was found to be a biomarker for predicting severe COVID-19 in Japan by Sugiyama et al. IFN-λ3 upregulates PAMP-mediated production of proinflammatory cytokines that induce high expression of inflammatory mediators such as IL-6, IL-12, and TNF. It has an association with neutrophil degranulation, complement, and coagulation cascades [19]. In the study by Sugiyama et al. [12], they tested the predictive ability of oxygen administration by the values of INF-λ3 and TARC at the time of hospitalization, but the timing of patient hospitalization was inconsistent. For example, despite their severity, all patients were hospitalized for isolation during the early pandemic. Therefore, it was unclear at which timing INF-λ3 or TARC should be used. Because effective monoclonal antibody therapy and oral antivirals are becoming available as treatment options for patients with mild disease [20–22], it would be useful to predict the disease severity at an early stage. IFN-λ3, which predicted future oxygen demand, may be a useful marker to determine whether their administration is suitable. IFN-λ3 increases early during the disease course and decreases quickly [12]. It is also unclear to what extent IFN-λ3 is useful in situations where oxygen administration has already been initiated.

Although it was not significant compared with other markers, it remained potentially useful as a predictor of the need for advanced oxygen devices such as NHFC. However, IFN-λ3 and TARC have not been validated in other countries and further studies are needed.

In terms of the other markers to predict disease severity, IL-6 plays a major role in "cytokine storms" and has the most validated data [2]. The association between IL-6 and disease severity has been noted since the early pandemic [23]. Anti-IL-6 agents are a therapeutic option and some reports have examined them as biomarkers of therapeutic efficacy [24]. In a meta-analysis, AUC was 0.875 to predict severe disease, but 0.531 to predict death, which was insufficient for prediction. However, the cutoff value was not established in each study. Although 42 pg/mL was used in this study, the cutoff values were assumed to be 12.7 pg/mL for oxygen administration prediction and 76.0 pg/mL for ventilator prediction. It is considered necessary to verify the cutoff in accordance with the required prediction. Although this study did not show any significant predictive ability, anti-IL-6 agents decrease IL-6 [25], and a small number of patients in Study 2 had received tocilizumab and baricitinib, inhibiting JAK upstream in the cascade, which may have some effect. However, although these agents are used for ill patients who require oxygen devices above NHFC, IL-6 did not differ significantly from oxygen devices above the NHFC in a secondary validation in which these subjects were excluded. Therefore, the effect of these agents was considered to be small.

CP has been proposed to be associated with COVID-19 severity [13] and its prognostic value has been verified in many reports with small sample sizes [26–33]. In a meta-analysis of severity prediction, there was a significant difference between severe and non-severe cases [34], but there are also reports that deny the utility of CP [27]. In the present study, we found a certain level of performance in predicting the need for oxygen administration and a mechanical ventilator, but its predictive ability was not as high as that of other biomarkers. Although serum was used in this study, it has been reported that AUC tends to be higher in plasma with EDTA than in serum. However, the difference does not appear to be significant [26]. There was a good correlation with immunochromatography for CP (S1 Fig), which may be of value as a POCT.

We used the simple risk score generated using Japanese data [17] and the PREDI-CO score [14] as a prognostic score to contrast with biomarkers. Other predictive scores (e.g., COVID-GRAM and NEWS2) that incorporate multiple factors, including common biomarkers such as CRP, LDH, and lymphocyte counts, have also been reported [14–16, 35, 36]. Although the predictive ability of Yamada et al. in the present two scores was not high, the results of the present study were reasonable because the cutoff was determined to increase sensitivity. PREDI-CO also showed not good predictive ability in predicting future oxygen demand, but it was good for critical cases. Several reports have investigated the prognosis prediction using combinations of multiple biomarkers [7, 12, 37, 38], which may improve the predictive power in this study. Using the scoring of the number of biomarkers positive at the cutoff calculated by the ROC curve for IL-6, IFN-λ3, TARC, and CP, which predicted future mechanical ventilation demand, had a high AUC (0.86). IFN-λ3 performed well in predicting progression before oxygen administration, but none of the single biomarkers performed well in predicting progression after oxygen administration. Because exacerbation of the respiratory status after oxygen administration can be caused by various factors, including a cytokine storm, lung fibrosis, and bacterial infection, and it was speculated that a variety of biomarkers may be involved, progression may be difficult to predict by a single biomarker. It would be helpful to verify which biomarkers contribute to the prediction of exacerbations and to which extent. However, because the sample size was not large enough to weigh the results by regression analysis, it is necessary to re-evaluate the results with a sufficient sample size.

## Limitations

This study had two major limitations. The first is the sample size. The frequency of blood sampling for patients without oxygen demand was low, and most patients admitted to hospital have an oxygen demand. Therefore, the sample size for prediction was very small, especially for patients without oxygen administration. The inclusion of these patients in a prospective study may be limited because the diagnosis is often confirmed after the patient returns home. Additionally, not all patients within the time period were included, and the inclusion was arbitrary in accordance with the disease severity. After inclusion, anonymization was performed, so that cases were not arbitrarily selected based on test results, but this does not indicate that there was no influence. However, for IFN-λ3 and TARC, a similar trend to the present study was observed in a larger sample size validation of a report that predicted the occurrence of future oxygen demand (Prediction A) in the early disease stages when oxygen administration had not started [39]. Prediction B, which predicts whether patients on oxygen will become more severely ill, was not demonstrated in this other report [39], and future studies should include this prediction. Outcome studies are also expected to reveal how many patients actually avoid hospitalization using biomarkers.

Second, the study period to incorporate the subjects was relatively long, resulting in a situation where infected persons of wild, alpha, and delta variants were mixed together as a pandemic in Japan. Additionally, in terms of the time period employed, the majority of the subjects were unvaccinated, but information on vaccination was not collected. The predictive power of biomarkers by vaccination and each variant has not been verified. Although there are no reports of these validations, we could not rule out an effect of these factors on each biomarker because the disease severity would be different. Thus, these should be shown in future studies.

## Conclusions

In this study, we used conserved serum to predict disease severity by considering the timing of COVID-19 sample collection, but we did not find any biomarker that showed a significant predictive ability. However, IFN-λ3 performed well as a predictor of future oxygen administration in subjects who were not receiving oxygen. Evaluation of combinations of the four biomarkers may be useful to predict the need for future mechanical ventilation.

## Supporting information

**S1 Appendix. Biomarker measurement methods.**
(DOCX)

**S1 Fig. Correlation between ELISA and absorbance immunochromatography readings for calprotectin.**
(TIF)

**S1 Table. Multiple comparisons of the AUC of ROC curves (p-values).** A: For prediction of future oxygen administration. B: For prediction of future mechanical ventilation administration or death. C: For prediction of future oxygen support with nasal high flow cannula or death.
(DOCX)

## Acknowledgments

We thank all staff at the Travel Clinic, Disease Control and Prevention Center, National Center for Global Health and Medicine, for collecting the clinical data. We thank Yumiko Kito

and Azusa Kamikawa for technical assistance. We also thank Mitchell Arico from Edanz (https://jp.edanz.com/ac) for editing a draft of this manuscript.

## Author Contributions

**Conceptualization:** Kei Yamamoto, Satoru Nebuya, Masato Kurokawa.

**Data curation:** Kei Yamamoto, Yusuke Ohsiro, Michiyo Suzuki, Sayaka Miura, Maki Nagashima, Satoru Nebuya.

**Formal analysis:** Kei Yamamoto.

**Funding acquisition:** Kei Yamamoto.

**Investigation:** Yusuke Ohsiro, Junko S. Takeuchi, Satoru Nebuya.

**Methodology:** Kei Yamamoto, Yusuke Ohsiro, Satoru Nebuya.

**Project administration:** Kei Yamamoto, Moto Kimura, Masato Kurokawa.

**Resources:** Sayaka Miura, Maki Nagashima, Noriko Iwamoto.

**Supervision:** Wataru Sugiura, Norio Ohmagari.

**Visualization:** Kei Yamamoto.

**Writing – original draft:** Kei Yamamoto.

**Writing – review & editing:** Kei Yamamoto, Yusuke Ohsiro, Tetsuya Suzuki, Junko S. Takeuchi, Moto Kimura, Wataru Sugiura, Satoru Nebuya, Masato Kurokawa, Norio Ohmagari.

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
