## [Decision Letter · Decision Letter 0]

28 Jul 2022

PONE-D-22-08807Validation of the severe COVID-19 prognostic value of serum IL-6, IFNλ3, CCL17, and calprotectin considering the timing of clinical need for predictionPLOS ONE

Dear Dr. Yamamoto,

Thank you for submitting your manuscript to PLOS ONE. After careful consideration, we feel that it has merit but does not fully meet PLOS ONE’s publication criteria as it currently stands. Therefore, we invite you to submit a revised version of the manuscript that addresses the points raised during the review process.

Please revise.

We look forward to receiving your revised manuscript.

Kind regards,

Academic Editor

PLOS ONE

Journal Requirements:

Reviewers' comments:

Reviewer's Responses to Questions

**Comments to the Author**

1. Is the manuscript technically sound, and do the data support the conclusions?

Reviewer #1: Partly

Reviewer #2: Yes

2. Has the statistical analysis been performed appropriately and rigorously? 

Reviewer #1: No

Reviewer #2: Yes

3. Have the authors made all data underlying the findings in their manuscript fully available?

Reviewer #1: No

Reviewer #2: Yes

4. Is the manuscript presented in an intelligible fashion and written in standard English?

Reviewer #1: Yes

Reviewer #2: Yes

5. Review Comments to the Author

Reviewer #1: Dear authors

Thanks for the interesting study.

The study design was impressed and valuable, but with major limitation, such as too small size.

Otherwise, beside study population size, I suggest that you could add outcome study in the future.

Reviewer #2: The authors submitted a research article in which they examined the ability of four biomarkers to predict disease severity using conserved sera collected at the appropriate time for prediction. They enrolled 18 and 45 hospitalized patients aged 20 years or older in study 1 and 2, respectively. Since then they collected blood samples, measured biomarkers, calculated the PREDI-CO score by the severity prediction model extracted from the above data and the simple risk score by the oxygen administration prediction model created from registry data in Japan. The authors found that the prediction of mechanical ventilator management was similar for IL-6, IFN-λ3, TARC, and CP biomarkers, and the PREDI-CO score. They concluded that there were no any biomarker that showed a significant predictive ability. The aim of the study is clear and concise. The manuscript has a logical structure and well-balansed subsections that cover all aspects of the initial hypothesis. The tables and figures are clear and legible. The conclusive part seems to show informative description of the finding significance. Although the study has a small sample size and retrspective design, it appears to be intriguing, because it confirms a role of conventional score to stratify the patients at risk. However, I would like to put forward several comments to discuss.

1. Whether the authors pooled the biomarkers fully related to all aspects of the natural evilution of the disease ?

2. Is it reasonable to identify different biomarkers at different pathologial stages of the disease ?

3. Please, give extensive explanation of the main reason why the biomarker model was not became effective .

6. PLOS authors have the option to publish the peer review history of their article (what does this mean?). If published, this will include your full peer review and any attached files.

Reviewer #1: No

Reviewer #2: No

---

## [Author Response · Author response to Decision Letter 0]

7 Aug 2022

Reviewer #1: 

Dear authors

Thanks for the interesting study.

The study design was impressed and valuable, but with major limitation, such as too small size.

Otherwise, beside study population size, I suggest that you could add outcome study in the future.

Thank you for your comments. We are aware that the sample size is a major limitation. However, during the review process, another research group, to which some of the coauthors belong, published a study with a slightly larger sample size (Suzuki T, et al. Eur J Clin Invest 2022), which showed data that supported the “Prediction A” results for IFNλ3 and TARC. Therefore, we have included this in the “Limitations” section. However, there were only eight patients who required advanced respiratory support beyond high-flow oxygen therapy in this other study and they were not receiving oxygen. Therefore, “Prediction B” still needs to be validated.

As suggested by the reviewer, we have added a perspective on outcome studies to determine whether patients were able to continue without hospitalization using the biomarkers.

In response to these points, the " However, for IFN-λ3 and TARC, a similar trend to the present study was observed in a larger sample size validation of a report that predicted the occurrence of future oxygen demand (Prediction A) in the early disease stages when oxygen administration had not started [40]. Prediction B, which predicts whether patients on oxygen will become more severely ill, was not demonstrated in this other report [40], and future studies should include this prediction. Outcome studies are also expected to reveal how many patients actually avoid hospitalization using biomarkers." was added in the Limitations section.

Revisions are highlighted in yellow in the text.

Reviewer #2: 

The authors submitted a research article in which they examined the ability of four biomarkers to predict disease severity using conserved sera collected at the appropriate time for prediction. They enrolled 18 and 45 hospitalized patients aged 20 years or older in study 1 and 2, respectively. Since then they collected blood samples, measured biomarkers, calculated the PREDI-CO score by the severity prediction model extracted from the above data and the simple risk score by the oxygen administration prediction model created from registry data in Japan. The authors found that the prediction of mechanical ventilator management was similar for IL-6, IFN-λ3, TARC, and CP biomarkers, and the PREDI-CO score. They concluded that there were no any biomarker that showed a significant predictive ability. The aim of the study is clear and concise. The manuscript has a logical structure and well-balanced subsections that cover all aspects of the initial hypothesis. The tables and figures are clear and legible. The conclusive part seems to show informative description of the finding significance. Although the study has a small sample size and retrospective design, it appears to be intriguing, because it confirms a role of conventional score to stratify the patients at risk. However, I would like to put forward several comments to discuss.

Thank you for your comments. We have addressed your comments below.

Revisions are highlighted in yellow in the text.

1. Whether the authors pooled the biomarkers fully related to all aspects of the natural evolution of the disease?

Although only some biomarkers were examined in this study, they are representative biomarkers that have been suggested to be involved in disease exacerbation in previous studies. However, direct comparisons with LDH and lymphocytes, which are common biomarkers associated with COVID-19 prognosis, are not possible. However, because these biomarkers are included in the PREDI-CO score, although the timing of evaluation is simplified to the time of admission, we believe that this is an indirect evaluation.

2. Is it reasonable to identify different biomarkers at different pathological stages of the disease?

Changing the biomarkers to be measured in detail for each stage of the pathological condition may not be practical in terms of some aspects, considering the cost and management complexity involved in the testing system. However, if the purpose of the medical facility is clear, we believe it is worthwhile. For example, predicting future oxygenation disorders in a clinic that mainly provides initial medical care or predicting more serious conditions (e.g., high-flow oxygen therapy and ventilatory management) in a hospital that does not provide advanced medical care.

3. Please, give extensive explanation of the main reason why the biomarker model was not became effective.

Although we cannot draw definite conclusions at this stage, a study with a slightly larger sample size from another research group, to which some of the coauthors belong, showed data that support the “Prediction A” results for IFN-λ3 and TARC. It is presumed that IFN-λ3 has a high predictive ability for future oxygen demand in the early disease stages when oxygen may not be required. However, when more severe conditions are observed, i.e., when intubation and ventilation are required, a combination of exacerbating factors may occur, such as a secondary bacterial infection and cytokine storm. Therefore, we believe that each biomarker may have similar predictive performance. Therefore, the prediction of disease severity using more than one biomarker had a higher AUC. Although it was not the primary outcome of this study, we believe that the use of multiple biomarkers should be considered when predicting recurrence of severe disease. We have included some discussion of these considerations in the Discussion section.

---

## [Decision Letter · Decision Letter 1]

22 Sep 2022

PONE-D-22-08807R1Validation of the severe COVID-19 prognostic value of serum IL-6, IFNλ3, CCL17, and calprotectin considering the timing of clinical need for predictionPLOS ONE

Dear Dr. Yamamoto,

Thank you for submitting your manuscript to PLOS ONE. After careful consideration, we feel that it has merit but does not fully meet PLOS ONE’s publication criteria as it currently stands. Therefore, we invite you to submit a revised version of the manuscript that addresses the points raised during the review process.

Please revise.

We look forward to receiving your revised manuscript.

Kind regards,

Academic Editor

PLOS ONE

Reviewers' comments:

Reviewer's Responses to Questions

**Comments to the Author**

1. If the authors have adequately addressed your comments raised in a previous round of review and you feel that this manuscript is now acceptable for publication, you may indicate that here to bypass the “Comments to the Author” section, enter your conflict of interest statement in the “Confidential to Editor” section, and submit your "Accept" recommendation.

Reviewer #1: All comments have been addressed

Reviewer #3: (No Response)

2. Is the manuscript technically sound, and do the data support the conclusions?

Reviewer #1: Yes

Reviewer #3: Partly

3. Has the statistical analysis been performed appropriately and rigorously? 

Reviewer #1: Yes

Reviewer #3: Yes

4. Have the authors made all data underlying the findings in their manuscript fully available?

Reviewer #1: Yes

Reviewer #3: Yes

5. Is the manuscript presented in an intelligible fashion and written in standard English?

Reviewer #1: Yes

Reviewer #3: Yes

6. Review Comments to the Author

Reviewer #1: Thanks for the authors' comprehensive response, I do not have any more questions!

I would like to suggest this manuscript be accepted for publication.

Reviewer #3: Dear editor,

The main aim/hypothesis of the draft paper is really crucial in our clinical practice.

However, I have major concerns listed below.

Abstract section:

1. “…collected at the appropriate time…”. What do the authors mean? They should be clear about appropriate time.

2. Statistical methods might be excluded from the abstract section. They should be stated in the text (Method section). Instead, the method (sampling time, etc) or the features of the cases should be stated in abstract. Understanding of the methodology is difficult from the abstract section. For instance, current study cases have COVID pneumonia or COVID-associated other organ involvements??? The definitions of prediction A and B are not clear as in Introduction part of the text.

3. In conclusion part of the abstract, authors stated a sentence as “….combination of biomarkers was predictive of mechanical ventilator requirements…..”. There was no any result to conclude this in result section of the abstract. Those results should be added as “Which combinations? 2 or 3 combinations?.. etc”.

4. In conclusion part of the abstract, small sample size is limitation of the text. It should be in the text as limitations.

Method section:

1. They should explain the components of PREDI-CO score. How do they define severity as high or low?

Result section:

1. In Table 1. The “high risk” mean is not clear according to the result section?

2. For “…median values of IL-6, IFN-λ3, TARC, and CP were 10.4 (4.0–20.9)…” The ranges show us (min-max) OR (IQR). As far as I understood it shows us IQR; however, it should be stated again in here.

3. There is confusion about Table numbers. I think, they mean Table 2, while they are stating Table 3 in the text.

4. For the cases included in Study 1 and Study 2, there might be a Flow chart (Figure x) to understand actual situation. For example, I could not understand whether 5 cases required ECMO are needed non-invasive positive pressure ventilation OR invasive mechanical ventilator OR NOT. Via flow chart, understanding will be easy.

5. There is no any statement for Figure 1 in the text.

6. The legend of Figure 2A and 2B should be detailed. It is not easily understandable in this way. Moreover, supplement figure 2 and figure 3 might be added to Figure 2 as 2C and 2D.

7. For ROC curves, while they are writing “significant”, is it possible to show “p values”?

8. Exact number of the cases treated with anti-IL 6 or other immunomodulatory should be stated in the text.

9. I could not see the mortality ratios for the cases included in Study 1 and 2. Should be stated.

Discussion section (conclusion part):

They should state the implementation need of a biomarker such as IFN-λ3.

Sincerely

7. PLOS authors have the option to publish the peer review history of their article (what does this mean?). If published, this will include your full peer review and any attached files.

Reviewer #1: No

Reviewer #3: **Yes: **Yasemin Ozsurekci

---

## [Author Response · Author response to Decision Letter 1]

5 Oct 2022

Reviewer #1: 

Thanks for the authors' comprehensive response, I do not have any more questions!

I would like to suggest this manuscript be accepted for publication.

Thank you for your constructive comments.

Reviewer #3: 

The main aim/hypothesis of the draft paper is really crucial in our clinical practice.

However, I have major concerns listed below. 

Thank you for your useful comments.

Our point–by-point responses are below. Revisions are highlighted in yellow.

Abstract section:

1.“…collected at the appropriate time…”. What do the authors mean? They should be clear about appropriate time.

Regarding the appropriate timing, the following text was added: “Here, we examined the ability of four biomarkers to predict disease severity using conserved sera from COVID-19 patients who received inpatient care between January 1, 2020 and September 21, 2021 at the National Center for Global Health and Medicine, collected at the appropriate time for prediction. We predicted illness severity in two situations: 1) prediction of future oxygen administration for patients without oxygen support within 8 days of onset (Study 1) and 2) prediction of future mechanical ventilation support (excluding non-invasive positive pressure ventilation) or death of patients within 4 days of the start of oxygen administration (Study 2).”

2.Statistical methods might be excluded from the abstract section. They should be stated in the text (Method section). Instead, the method (sampling time, etc) or the features of the cases should be stated in abstract. Understanding of the methodology is difficult from the abstract section. For instance, current study cases have COVID pneumonia or COVID-associated other organ involvements??? The definitions of prediction A and B are not clear as in Introduction part of the text.

Thank you for your comments. Although reviewer #3 recommended omitting the description of statistical analysis, we retained the first sentence “AUC was calculated and compared by ROC curves” because we believed that deleting it would make the results unclear. 

In the Abstract, “Prediction A” and “Prediction B” were changed to “Study 1” and “Study 2”, respectively, to align with the main text. Research methods have been added, including the study duration, methods to obtain clinical information, and comparison of the predictive ability.

The sentence “2) prediction of future oxygen administration within 4 days of the start of oxygen administration (excluding non-invasive positive pressure ventilation; prediction B)” was changed to “2) prediction of future mechanical ventilation support (excluding non-invasive positive pressure ventilation) or death of patients within 4 days of the start of oxygen administration (Study 2)” because it incorrectly stated what was to be considered. 

“Study 1 included 18 patients, five of whom had developed oxygen needs. Study 2 included 45 patients, 13 of whom required ventilator management or died” was also added to the manuscript. However, this report is not a case study and we believe that detailed patient features are not needed in the abstract.

3. In conclusion part of the abstract, authors stated a sentence as “….combination of biomarkers was predictive of mechanical ventilator requirements…..”. There was no any result to conclude this in result section of the abstract. Those results should be added as “Which combinations? 2 or 3 combinations?.. etc”.

The “the number of” statement appears to have been lost and has been corrected. Additionally, the sensitivity and specificity when two or more tests were positive have been described and changed as follows: “The number of biomarkers above the cutoff showed the possibility of good prediction with an AUC of 0.86 (95% CI 0.75–0.97). When two or more biomarkers were positive, sensitivity and specificity were 0.92 and 0.63, respectively. In terms of biomarker testing at times when prognostication may be clinically useful, IFN-λ3 was predictive of oxygenation demand and a combination of four biomarkers was predictive of mechanical ventilator requirement.”

4.In conclusion part of the abstract, small sample size is limitation of the text. It should be in the text as limitations.

We have removed this part of the text in response to your suggestion.

Method section:

1. They should explain the components of PREDI-CO score. How do they define severity as high or low?

PREDI-CO predicts severe respiratory failure and we have added the definition. We have also included the “high risk” definition to address the later point. “The PREDI-CO score predicts severe respiratory failure, which is defined by WHO criteria SpO2 <93% with 100% FiO2 (reservoir mask, continuous positive airway pressure ventilation, or other non-invasive ventilation), respiratory rate >30 breaths/minute, or respiratory distress. Three or higher predicts a high risk for severe disease.”

2.In Table 1. The “high risk” mean is not clear according to the result section?

We have added a note to the Methods regarding the fact that scoring was performed in accordance with the age group.

3.For “…median values of IL-6, IFN-λ3, TARC, and CP were 10.4 (4.0–20.9)…” The ranges show us (min-max) OR (IQR). As far as I understood it shows us IQR; however, it should be stated again in here.

In the statistical analysis section, we stated that “Discrete and continuous data are expressed as numbers (percentages) and medians [interquartile range (IQR)], respectively,” but we have added that all of them are IQR. In some places, 95% CI was used. Therefore, we have specified this clearly. The term "Discrete and continuous data" has been changed to "Qualitative and quantitative data" because the terminology was not accurate.

4. There is confusion about Table numbers. I think, they mean Table 2, while they are stating Table 3 in the text.

Thank you for pointing this out. We have corrected the citation to Table 2.

5.For the cases included in Study 1 and Study 2, there might be a Flow chart (Figure x) to understand actual situation. For example, I could not understand whether 5 cases required ECMO are needed non-invasive positive pressure ventilation OR invasive mechanical ventilator OR NOT. Via flow chart, understanding will be easy.

As reviewer #3 indicated, we have included a flowchart diagram summarizing the outcomes as Fig. 2.

6.There is no any statement for Figure 1 in the text.

The description has been included in the “Introduction.”

7.The legend of Figure 2A and 2B should be detailed. It is not easily understandable in this way. Moreover, supplement figure 2 and figure 3 might be added to Figure 2 as 2C and 2D.

Supplementary Figure 3 has been integrated as Fig. 3C and an explanation was added. If you still believe that the explanation is insufficient, we would appreciate it if you could point out specific areas that need to be explained. However, because Supplementary Figure 2 is difficult to integrate because of its title, we have added it as Fig. 4.

8.For ROC curves, while they are writing “significant”, is it possible to show “p values”?

The statistical analysis software EZR used does not calculate a p-value when calculating the AUC. We believe that whether the lower limit of the 95% confidence interval is greater than 0.5 is sufficient to determine whether the result is significant. However, using SPSS, the p-value can be calculated as asymptotic significance (two-tailed). The following table shows the results.

p value Calprotectin IFNλ3 IL-6 TARC CRP

Study 1 0.59 0.008 0.094 0.59 0.70

Study 2

To predict future mechanical ventilation administration or death 0.017 0.016 0.040 0.011 0.057

To predict future oxygen support above a nasal high flow cannula 0.239 0.002 0.19 0.056 0.46

However, the sentence “A significant difference was observed only for IFN-λ3” was deleted because it was redundant. Additionally, the p-value for the comparison of medians was added as follows “with significant differences only for IFN-λ3 (p=0.0017).”

9.Exact number of the cases treated with anti-IL 6 or other immunomodulatory should be stated in the text.

There were three patients and the following statement was added to the text: “Two patients were receiving an anti-IL-6 antibody (tocilizumab) and one patient was receiving a JAK inhibitor at the time of serum sampling.”

10.I could not see the mortality ratios for the cases included in Study 1 and 2. Should be stated.

They were no mortality in Study 1 and 15.6% (7/45) in Study 2, which was already listed in Study 2, but the numbers were incorrect, which have been corrected and the percentages added.

They should state the implementation need of a biomarker such as IFN-λ3.

Because there are many limitations and the disease severity changes drastically, we could not conclude from the results that the introduction of biomarkers should be emphasized. However, the utility of such a biomarker, if identified, was described in the “Introduction” section.

---

## [Decision Letter · Decision Letter 2]

19 Dec 2022

Validation of the severe COVID-19 prognostic value of serum IL-6, IFNλ3, CCL17, and calprotectin considering the timing of clinical need for prediction

PONE-D-22-08807R2

Dear Dr. Yamamoto,

We’re pleased to inform you that your manuscript has been judged scientifically suitable for publication and will be formally accepted for publication once it meets all outstanding technical requirements.

Kind regards,

Academic Editor

PLOS ONE

Additional Editor Comments (optional):

Reviewers' comments:

Reviewer's Responses to Questions

**Comments to the Author**

1. If the authors have adequately addressed your comments raised in a previous round of review and you feel that this manuscript is now acceptable for publication, you may indicate that here to bypass the “Comments to the Author” section, enter your conflict of interest statement in the “Confidential to Editor” section, and submit your "Accept" recommendation.

Reviewer #1: All comments have been addressed

2. Is the manuscript technically sound, and do the data support the conclusions?

Reviewer #1: Yes

3. Has the statistical analysis been performed appropriately and rigorously? 

Reviewer #1: Yes

4. Have the authors made all data underlying the findings in their manuscript fully available?

Reviewer #1: Yes

5. Is the manuscript presented in an intelligible fashion and written in standard English?

Reviewer #1: Yes

6. Review Comments to the Author

Reviewer #1: Thanks for the authors' comprehensive response. The authors have completely response my previous questions and suggestion.

7. PLOS authors have the option to publish the peer review history of their article (what does this mean?). If published, this will include your full peer review and any attached files.

Reviewer #1: No

---

## [Editor Report · Acceptance letter]

22 Mar 2023

PONE-D-22-08807R2 

Validation of the severe COVID-19 prognostic value of serum IL-6, IFN-λ3, CCL17, and calprotectin considering the timing of clinical need for prediction 

Dear Dr. Yamamoto:

I'm pleased to inform you that your manuscript has been deemed suitable for publication in PLOS ONE. Congratulations! Your manuscript is now with our production department. 

Kind regards, 

on behalf of

Dr. Robert Jeenchen Chen 

Academic Editor

PLOS ONE